# Study of the Filtration Performance of Multilayer and Multiscale Fibrous Structures

**DOI:** 10.3390/ma14237147

**Published:** 2021-11-24

**Authors:** Vânia Pais, Carlos Mota, João Bessa, José Guilherme Dias, Fernando Cunha, Raul Fangueiro

**Affiliations:** 1Fibrenamics, Institute of Innovation on Fiber-based Materials and Composites, University of Minho, 4800 Guimarães, Portugal; cmota@tecminho.uminho.pt (C.M.); joaobessa@fibrenamics.com (J.B.); fernandocunha@det.uminho.pt (F.C.); rfangueiro@dem.uminho.pt (R.F.); 2Centre for Textile Science and Technology (2C2T), University of Minho, 4800 Guimarães, Portugal; 3Poleva—Termoconformados, S.A. Rua da Estrada, 4610 Felgueiras, Portugal; josedias@poleva.pt; 4Department of Mechanical Engineering, University of Minho, 4800 Guimarães, Portugal

**Keywords:** electrospinning, nanofibres, filtration, particles retention, multilayer systems

## Abstract

As the incidence of small-diameter particles in the air has increased in recent decades, the development of efficient filtration systems is both urgent and necessary. Nanotechnology, more precisely, electrospun nanofibres, has been identified as a potential solution for this issue, since it allows for the production of membranes with high rates of fibres per unit area, increasing the probability of nanoparticle collision and consequent retention. In the present study, the electrospinning technique of polyamide nanofibre production was optimized with the variation of parameters such as polymer concentration, flow rate and needle diameter. The optimized polyamide nanofibres were combined with polypropylene and polyester microfibres to construct a multilayer and multiscale system with an increased filtration efficiency. We observed that the penetration value of the multilayer system with a PA membrane in the composition, produced for 20 min in the electrospinning, is 2.7 times smaller than the penetration value of the system with the absence of micro and nano fibers.

## 1. Introduction

The incidence of particles with a small diameter (lower than 2.5 µm) in the air has risen in recent decades [1]. The rapid growth of urbanization and industrialization has led to a release of small particles to the atmosphere, such as solid particles and liquid droplets, which is concerning. Particles with diameters smaller than 2.5 µm may cause considerable damage due to their ability to penetrate the human bronchi, lungs and even the extrapulmonary organs. Furthermore, these particles can be linked to bacteria or viruses and cause serious human health problems due to the development of acute and chronic diseases [2,3]. Developing a solution to this problem is extremely important, and certain approaches such as filtration membranes have been identified as useful. These filter membranes can be applied in various products, such as face masks and NBC suits (protection against nuclear, biological, and chemical warfare agents) [4].

There are several types of particles that require filtering and each one of them has unique properties. The particle’s diameter is one such property, and it can span from a few nanometers, as is the case for antibodies and viruses, to microns, such as for pollen [5]. Figure 1 shows the previously mentioned variation in particle size retention. Therefore, since there are different particles with distinct sizes, the filtration process should be optimized, depending on the objectives. Permeability, filtration performance and the uniformity of the structure are the three principal factors to consider when developing or applying a filtration process. Permeability, mostly related to breathability and water vapour transmission, should be optimized to make the structure wearable without compromising the filtration efficiency [4]. Concerning the filtration performance, the filtration theory provides that the efficiency of this process increases with a decrease in the dimension of the fibres that compose the filter. This statement relates to the increase in fibres per unit area, which leads to an increase in the probability of impact between the filter and particles that need to be filtered. The uniformity is related to efficacy [6].

The filtration process occurs through different methods, dependent on the size of the particle requires filtering, and the parameter that most influences this. In Table 1 the different mechanisms of filtration are described, corresponding with the particles size being filtered. Larger particles are usually trapped by gravity sedimentation, since the pore sizes of filters are smaller than the particles size, blocking the particles outside the porous structure. Inertial impaction is also a possibility in the retention of larger particles. This mechanism occurs when the particles do not follow the direction of the airflow due to their large inertia. Thus, when associating high speed with larger particles, there is an increase in the probability of collision between particles and fibres. After collision, the particles can adhere to the fibres, and are retained in the filter. The interception mechanism is related to the retention of particles below 0.6 µm and occurs when the particles follow the airflow. Eventually, the particles come into contact with fibres that compose the filter and remain connected by Brownian forces. The efficiency of this process increases with the decrease in particle size. For nano-sized particles (below 0.2 µm), diffusion is the predominant mechanism. The particles do not follow the streamline direction and have a very slow and random movement. At some point, the particles and fibres collide and remain attached. Electrostatic attraction occurs in particles of different dimensions and occurs when the fibres are electrically charged and can capture the particles that are oppositely charged [5,7,8].

Concerning the retention of small particles, diffusion is the predominant mechanism of filtration. The diffusion efficiency increases with a decrease in the diameter of the fibres that constitute the filter. However, when the particles have a size of around 0.3 µm, the retention process can be harder to achieve because the diffusion mechanism may not occur. Therefore, to increase retention via interception, a multi-layer approach should be applied [5]. To promote nanoparticles’ retention, two factors should be considered: the use of fibres with very small diameters in the filter membrane and the application of a multi-layer system [9,10,11].

Concerning the relationship between fibres with lower diameters and higher efficiencies, nanofibres have been identified as a solution with great potential. Nanofibre filters have a controllable small diameter, low basis weight, high permeability values, reduced thickness and a porous structure [7,12]. The electrospinning technique is a simple and effective method used to produce fibres at a nanoscale. In this methodology, a high electric field is applied which promotes repulsive interactions among the polymeric solution, and the Taylor cone is formed. When the electrostatic forces overlap the repulsive interactions, a charged jet is ejected from the Taylor cone with a dynamic whipping. Concurrently, the solvent evaporates, and the jet is stretched into fibres with finer diameters that are deposited on a grounded collector [13,14]. The electrospun nanofibre diameters can range from a few nanometers to micrometres. Additionally, this filtration layer can be produced with distinct raw materials and by the application of different parameters, obtaining specific nanofibres with several functionalities. So, the nanoscale diameters and the interconnected porous structure of the electrospun nanofibres make the electrospinning technique a very attractive approach for filtration applications. Furthermore, the static charge, which is a result of the electrospinning process, may remain on the fibres that have been produced and enhance the filtration retention by electrostatic attraction [12,15,16].

The selection of the most suitable polymer to produce the fibres to be applied in particles retention should consider good mechanical properties, hydrophobicity, biocompatibility and compatibility with non-toxic solvents [17,18,19]. Polyamide (PA) stands out as a potential polymer since it has excellent chemical stability and thermal resistance. It is a synthetic polymer that is biodegradable and biocompatible. Usually, polyamide is dissolved in formic acid, and this combination can be electrospun to efficiently produce nanometric fibres [17,18], as opposed to polycaprolactone (PCL) (for example), which is usually dissolved in chloroform and dimethylformamide (DMF) [11]. According to EU directive 67/548/EEC, DMF is toxic [18].

In this study, electrospun nanofibres were produced with PA polymers to optimize systems with a higher filtration efficiency. Firstly, the parameters’ polymers concentration, flow rate and needle’s diameter were optimized to obtain fibres with very low diameters and a mat with controlled porosity. The morphology and intrinsic properties of the produced nanofibres were analyzed. In the second part of the study, several combinations of the optimized electrospun PA nanofibres with polypropylene (PP) and polyester (PES) microfibres were analyzed to obtain a multi-layer and multiscale system with a high retention capacity of small particles. The performance of the obtained combinations was evaluated by measuring the filtering material penetration, air permeability and breathing resistance.

## 2. Materials and Methods

### 2.1. Materials

PA 6.6 pellets (with a molecular weight of 262.35 g/mol, Tm = 250–260 °C, density = 1.14 g/mL at 25 °C, Sigma Aldrich, St. Louis, MO, USA) were used as a polymeric matrix. The solvent used was formic acid (FA) (98–100%, Fisher Scientific, Leics, UK).

The PP microfibres membrane (weight = 50 g per square meter (gsm), thickness = 360 µm, average fibre diameter = 3.7 µm), applied as a substrate for PA nanofibres, were obtained from Protechnic S.A. (Cernay, France).

### 2.2. Production of Electrospun PA Membranes

The polymeric solution was optimized after studying different PA concentrations (20% *w/v* and 25% *w*/*v*) to obtain fibres with small diameters and without defects. The solvent applied was FA in 1:1 proportion. The polymeric solution was prepared through the dissolution of PA pellets in FA, for at least 6 h at 30 °C, at constant stirring.

PA nanofibre webs were produced by electrospinning NF-103 from MECC Co., Ltd. (Fukuoka, Japan). The electrospinning parameters were also optimized to obtain fibres with small diameters to increase filtration efficacy. The tested parameters include the flow rate (ranging from 0.4 to 2 mL/h), needle’s diameter (0.33, 0.41 and 0.61 mm) and the fibre deposition time (10, 20 and 30 min). The voltage applied was 28 kV and the collector-needle distance was 100 mm. The electrospinning process was conducted at 60% ± 5 RH and 20 °C ± 2. The group’s previous studies were consulted to define certain fixed parameters, including voltage, collector-needle distance and the applied solvent [20,21]. The studied conditions and the corresponding produced membranes are presented in Table 2.

After optimizing the ideal conditions to produce the nanofibres, deposition was performed over PP microfibres to construct a multilayer and multiscale system with higher performance in terms of small particles retention. Another layer of PP microfibres were added to the PA nanofibres deposited above the PP microfibres. Thus, the filtration layer is composed of 3 layers: PP microfibres, PA nanofibres and PP microfibres. Two different polyester (PES) nonwovens were added to the filtration layer, one for the inner layer—PES IL—and the other for the outer layer—PES OL. A schematic representation of the multilayer system is presented in Figure 2. A total of 4 different combinations were obtained, as represented in Table 3. The difference between the multiple combinations relates to the nanofibre production time, of 0, 10, 20 and 30 min. The membranes were combined through a thermoforming process using a mould with a face-mask shape. However, it is important to mention that the structure can be molded to other shapes for application in other types of products.

### 2.3. Electrospun PA Membranes Characterization

The morphology of the produced PA nanofibres was investigated using a scanning electron microscope (SEM). The analyses were performed using a NOVA 200 Nano SEM from the FEI Company (Hillsboro, OR, USA). Due to the polymeric nature of the analysed specimens, the samples were vacuum metalized with a thin film of gold-palladium (Au-Pd) before the analysis. The average diameters and the porous distribution of the fibres were calculated by taking measurements from different regions using ImageJ software (1.52a).

The chemical composition and the structural aspects of the electrospun nanofibres were analyzed using Fourier transform infrared spectroscopy (FTIR) coupled with the attenuated reflection (ATR) technique using IRAffinity-1S, SHIMADZU equipment (Kyoto, Japan). All spectra were obtained in the transmittance mode with 45 scans over a wavenumber range of 4000–400 cm^−1^.

The thermal behaviour of the electrospun nanofibres was assessed via a thermogravimetric analysis (TGA) of the electrospun fibres, which was performed with an STA 700 from HITACHI (Tokyo, Japan). The samples were tested in a temperature range from 25 °C to 500 °C, at a heating rate of 10 °C/min.

### 2.4. Electrospun PA + Nonwoven Fabrics Characterization

The PP microfibres, PES IL and PES OL purchased from Protechnic S.A. (Cernay, France) were also studied. Morphology was analyzed through brightfield microscopy using a Microscope Leica DM750 M (brightfield) (Leica, Wetzlar, Germany) with a coupled camera.

The areal mass of several layers from the multilayer system was calculated by dividing the weight of the sample mat by its effective area. The thicknesses of the different membranes and multilayers systems were measured by analyzing the cross-section at the corresponding SEM images. The areal mass and thickness were obtained by performing measurements on 10 different regions of the samples.

The air permeability, filtration and respiratory evaluations were performed according to the standard EN 149:2001+A1:2009., For the air permeability evaluation, 40 Pa pressure was applied, using an air permeability tester from TEXTEST instruments (Zurich, Switzerland), model FX 3300. The tests related to filtration and respiratory evaluation were performed at Aitex—textile research institute, in Spain. The penetration of sodium chloride aerosol was tested by applying a flow rate of 95 L/min, and the maximum value was registered after 3.5 min of exposure. To perform the respiratory resistance evaluation, the pressure at 3 different conditions was registered, namely, inhalation at 30 L/min, inhalation at 95 L/min and exhalation at 160 L/min.

## 3. Results

### 3.1. Electrospun Fibre Characterization

#### 3.1.1. Fourier Transform Infrared Spectroscopy and Thermogravimetric Analysis

The PA nanofibres were produced using an electrospinning technique according to the conditions reported in Table 2. The FTIR spectra of the nanofibres produced with PA polymer are shown in Figure 3. From the figure, it is possible to identify one significant band at 3300 cm^−1^, typically attributed to the N-H stretching vibration. The asymmetric and symmetric stretching of CH_2_ appears at 2931 and 2860 cm^−1^, respectively. In this spectrum, amide bands of the PA 6.6. appear at 1637 and 1537 cm^−1^, and the bands located at 1145 cm^−1^ correspond to the CO–CH symmetric bending vibration when combined with CH_2_ twisting. The bands at 935 and 688 cm^−1^ are typically attributed to the stretching and bending vibrations of C–C bonds and the band at 580 cm^−1^ may be a result of O=C–N bending. The FTIR spectra obtained agree with those of other authors as a PA 6.6. FTIR spectrum [22,23,24,25].

#### 3.1.2. Thermogravimetric Analysis

To study the polymer behaviour at different temperatures, a TGA analysis was performed. The results are represented in Figure 4. At the TGA plot, a rapid decline in values starting at 350 °C is visible. No significant changes are observed preceding this temperature value. In this way, the PA polymer is found to have a very resistant thermal profile, that is only able to support temperatures up to approximately 320 °C. After this, a single-step degradation of the electrospun fibres is visible. The obtained results are in accordance with the results of other authors [26,27,28].

#### 3.1.3. Morphological Analysis

The PA nanofibre morphology was studied using SEM images. The images and the corresponding fibre diameters and porous distribution are represented in Table 4.

The SEM images reveal that randomly deposited fibres were produced. This feature is important, as filtration aims to produce a structure composed of innumerable pores with very small dimensions. This way, the probability of promoting the retention of nanoparticles is enhanced. The porous distribution, based on size, was measured using imageJ software. The obtained values ranged from 21 to 56%. To promote higher filtration efficiencies, a perfect combination of the fibres diameter and porous distribution should be achieved—a small fibre diameter allows for a lot of tinny pores, but the percentual distribution based on the size of these pores should be as small as possible, to increase the retention of particles probability. The PA nanofibres with the highest polymer concentration have the highest value of porous distribution. Most of the samples obtained percentage distribution values varying from 20 to 30%, thus emphasizing the applicability of PA nanofibres in the field of filtration. It is also important to produce fibre matts with higher densities, to ensure an increase in the collision points’ probability. For this purpose, samples A, D, E, F, G and H are identified as viable options. Regarding the diameters of the fibres, samples A and D were found to be significant. Both samples were produced with a flow rate of 0.4 mL/h, with a needle with a diameter of 0.33 mm applied on sample A, and a needle with a diameter of 0.41 mm applied on sample B. Although the two samples have the two lowest mean fibre diameters, sample B had a smaller value—293 nm. Furthermore, samples A and D also have the lowest standard deviation value, which means that the produced fibres are more uniform with each other in terms of size. Sample I was produced using the same ideal conditions as sample B, however, the polymer proportion was higher (25% *w/v* instead of 20% *w*/*v*). When comparing sample I with the others, it is clear that the number of fibres produced per unit area was considerably lower. The average diameters of the produced fibres, as well as the corresponding standard deviation, are higher when compared to the other samples. This result reveals that the application of 20% (*w*/*v*) PA is the best approach to produce homogeneous mats of PA nanofibres with smaller diameters.

It can be observed that different parameters in PA-nanofiber production affect the fibre’s diameter and porous distribution values, as presented in Figure 5. The values are related to the samples produced with 20% (*w*/*v*) PA. By analyzing the bars related to the diameter of the needles between 0.33 mm and 0.41 mm in Figure 5a, it is possible to conclude that nanofibres with lower diameters are always obtained when using a needle with a diameter equal to 0.41 mm. Additionally, by analyzing the flow rate values, it is possible to conclude that generally, lower flow rates promote the production of fibres with smaller diameters. The results of porous distribution are presented in Figure 5b, and it seems that a direct correlation between this value and the nanofiber construction parameters cannot be established. However, lower values for porous distribution are related to smaller fibres diameters, which becomes apparent when looking at both Figure 5a,b. When combining lower flow rates with smaller needle diameters, homogeneous mats of nanofibres (low standard deviation values) with small diameters and a porous distribution are produced. These conditions are optimal filtration purposes [6].

### 3.2. Electrospun PA + Non-Woven Fabrics Characterization

The above-mentioned optimal conditions for producing PA nanofibres, using electrospinning methods, were applied in this phase of the study The production and deposition of PA nanofibres onto the PP microfibres were conducted to obtain a multilayer structure for filtration purposes. The PP microfibres were morphologically and physically characterized, and are presented in the microscope images in Figure 6. PP fibres were found to have a random orientation. The average fibre diameter is 3.7 µm (Table 5). This value is much higher than the value obtained for PA nanofibres. Due to the conjugation of these 2 different structures, a multiscale system was constructed, and the filtration efficiency is expected to increase with minimal levels of breathability and/or permeability. The nanofibre membrane is much thinner (ranges from 1.16 to 2.64 µm depending on the production time for PA nanofibres) than the microfibres membrane (368.3 µm), making its application on wearable devices much more attractive. This is one of the greatest advantages of nanoscale, through the addition of a very small amount of nanomaterials it is possible to maximize specific functionalities on a high scale.

PA electrospun nanofibres were produced and deposited succesfully. In total, 3 different combinations were obtained, since 3 different deposition times of the PA nanofibres were studied: 10, 20 and 30 min. A control sample consisting of the PP membrane without PA nanofibres was added to the study. By changing the time of nanofibre production, a different aerial mass, thickness and air permeability were obtained. These differences will influence the filtration efficiency. Results are shown in Figure 7. All the parameters corresponded to increasesin the of PA nanofibre production time. In the case of aerial mass results, it is observed that the value increased with a higher deposition time of PA nanofibres. This result was expected since higher deposition times allow for the production of more nanofibres, thereby increasing its density. The sample in which nanofibres were produced for 10 min is very similar to the PP microfibres. The sample in which nanofibres were produced for 20 min has an aerial mass closer to the sample in which nanofibres were produced for 30 min. This result indicates that the deposition time is not uniform, as differences were notices with regard to time. This relationship between higher deposition times and the properties of the produced electrospun nanofibres were also observed by Vrieze et al. The authors stated that after some time of fibre deposition the electric field becomes distorted, causing an inhomogeneous deposition in the collector [29]. The thickness of the samples are very similar to each other. This result is a consequence of the higher thickness of the PP microfibres (368.3 µm) compared to the PA nanofibres (10 min: 1.16 µm; 20 min: 1.52 µm; 30 min: 2.64 µm). So, when combining the structures, the PA nanofibres thickness is hidden by the PP microfibres due to the different thicknesses of the two structures. It was also observed that the PP microfibres have a non-homogeneous mat in terms of thickness, with a high standard deviation value. It is also possible to conclude that the addition of PA nanofibres decreases the permeability values, due to the presence of extra material. Once again, greater differences are observed between the samples for which nanofibres were produced for 10 min and 20 min, and the samples produced during 20 and 30 min. The air-permeability parameter is more affected by the presence of the nanofibre membrane. On one hand, lower air permeability values may increase the filtration efficiency, however, on the other hand, it may compromise the breathability parameter. So, this topic needs to be considered when analyzing the filtration parameter and some optimization may be required to obtain the appropriate equilibrium between these two factors.

The SEM images shown in Figure 8 represents three different deposition times of the electrospun nanofibres onto the PP microfibres studied. Due to the different dimensions of PA (nano-scale) and PP (micro-scale) fibres, the two fibres are perfectly distinguishable from each other. From the images provided, it is possible to observe that two different layers were obtained via the establishment of contact points between the two adjacent layers. It is also clear that the presence of the PP membrane as a substrate during nanofibres production did not affect the electrospinning process, as the nanofibres with the presence of the PP membrane have the same morphology as the ones represented in Table 4. The density of the PA nanofibres increases with the deposition time, observable in the SEM images. This result increases the aerial mass and thickness and decreases the air permeability, as was observed in the results represented in Figure 7.

Since the multilayer filtration system is intended as personal protective equipment, it will probably have direct contact with skin on one side and with the external environment on the other side. For this reason, two extra layers were added, one as an inner layer and the other as an outer layer, to promote higher comfort levels and mechanical support, respectively. These two layers were composed of polyester (PES) polymer, for which microscopic images are represented in Figure 9. The PES IL is denser since it has been selected due to softness. The PES OL will be in contact with the environment and was chosen due to its higher rigidity to provide protection from possible dangers in the external environment and to support the entire structure. Therefore, a multilayer system with three different layers—inner layer (PES IL); filtration layer (PP microfibres + PA nanofibres + PP microfibres); outer layer (PES OL)—were defined. At this point, 4 different combinations were tested: the inner and outer layer remained the same throughout, the filtration layer changed because of the three different deposition times of the studied electrospun nanofibres. The several combinations studied are represented in Table 3. The membranes were combined using a thermoforming process and moulded using a mask shape mould.

The multilayer system has been optimized for application in filtration systems, to promote micro and nanoparticle retention. More specifically, it can be applied in products such as suits or respiratory masks. The filtration and respiratory parameters were analyzed in relation to the intended purpose of the filtration system. The results are represented in Table 6. The filtration topic was evaluated by the penetration of sodium chloride after 3.5 min of exposure, and it was observed that the multilayer system without the presence of PA nanofibres achieved much higher values (15.39%) than the systems with PA nanofibres (7.83, 5.9 and 7.11%). This result highlights that the presence of nanofibres promotes smaller penetration rates, and, consequently higher retention rates, thereby increasing the filtration efficiency. Therefore, the results show that the penetration value of the multilayer system, composed of a PA membrane, produced for 20 min, is 2.7 times lower than the penetration value of the system with the absence of PA nanofibres. It is also important to emphasize that the amount of PA present in the system is very small compared to the remaining layers that make up the entire system. By comparing the three multilayer systems, composed with PA nanofibres, the system composed of nanofibres produced for a smaller deposition time—10 min—was identified as having a higher rate of particle penetration and a lower efficiency in terms of filtration. The system in which nanofibres had been produced for 20 min resulted in smaller penetration rates with higher filtration capabilities. The results represented in Figure 7, related to aerial mass, thickness and air permeability reveal that the multilayer systems in which PA nanofibres were produced for 20 and 30 min are very similar. The deposition time can only be increased to a certain threshold, due to electric field distortion occuring in nanofibre production, thereby changing the deposition profile, an effect that has previously been described by Vrieze et al. [29]. So, the different values shown in Table 6 for the multilayer systems composed of nanofibres produced during 20 and 30 min are probably related to the lower homogeneity of the PA nanofibres produced for 30 min, creating points of lower density and decreasing the filtration efficiency.

The last three rows of Table 6 are related to the respiratory parameter and the evaluation is performed by analyzing the resistance in millibar (mbar). The lower the resistance values, the greater the breathing facility. The system without the presence of nanofibres has the smallest resistance values, both in terms of exhalation and in terms of inhalation. This result is logical since the combination has one missing layer compared to the other systems. Additionally, the higher value may not be problematic, if the values are under the limits defined by the legislation. These limit values change according to the location in which the product is distributed and also depending on the type of product. For example, in Europe, the legislation that is applied is the RfU PPE-R/02.075.02—certification of filtering half mask against SARS-CoV-2. This legislation defines that when analyzing the respiratory resistance of an FFP2 mask, in inhalation at 30 m/min, the maximum value permitted is 0.7 mbar and at 95 L/min the maximum value permitted is 2.4 mbar. Additionally, when evaluating the exhalation at 160 L/min the value must not exceed 3.0 mbar [30]. In the results obtained within this work, the respiratory resistance at inhalation achieved values that are under values specified by the legislation. In the exhalation evaluation, it was observed that the masks composed of nanofibres membranes produced during 20 and 30 min have a respiratory resistance value that is slightly higher than defined by legislation. Therefore, for this specific application, this aspect should be optimized. Optimization can be performed by changing the electrospinning parameters, such as the production time or the flow rate, or by changing the thermoforming parameters, such as the temperature or the time of forming. When comparing the three multilayers composed with PA nanofibres, very similar results are obtained to the three systems at each respiratory resistance condition. The value that differs the most is the exhalation evaluation at the multilayer system with PA nanofibres produced for 30 min. This value is slightly higher and is related to the presence of more material. At this point, the differences previously observed for 20 and 30 min are no longer observable because the respiratory parameters are affected by the entire structure.

## 4. Conclusions

Multilayer systems composed of PA nanofibres at the filtration layer were produced. Within this study, different polymer concentrations, flow rates and needle diameters, applied during electrospinning technique, were tested to define suitable conditions for filtration. It was concluded that lower flow rates, combined with smaller needle diameters, enhance the production of nanofibres with smaller diameters. In the second section of the study, different multilayer systems, with PA nanofibres located at the filtration layer, were studied. The PA nanofibres between the different multilayer systems had different production times which led to membranes with a distinct aerial mass, thickness and air permeability. The three-pointed properties affect the filtration efficiency. It was observed that the ideal deposition time for the PA nanofibres is 20 min. The nanofibres, together with the substrate layer—PP microfibres—have an aerial mass equal to 55.28 gsm, a thickness of 370.15 µm and an air permeability of 26.5 L/m^2^/s. For lower values, the filtration performance was compromised due to a lack of sufficient material to prevent the penetration of the particles. On the other hand, for higher time periods of nanofibre production, it was observed that the properties of the samples do not increase proportionally because the produced nanofibre mats suffer some distortion during the electrospinning technique. The most important conclusion taken from this study is the high potential of nanotechnology, more precisely of electrospun nanofibres, since, with the addition of a very tinny layer of nanofibres, the penetration values decreased 2.7 times when compared with membranes without electrospun nanofibres.

## Figures and Tables

**Figure 1 materials-14-07147-f001:**
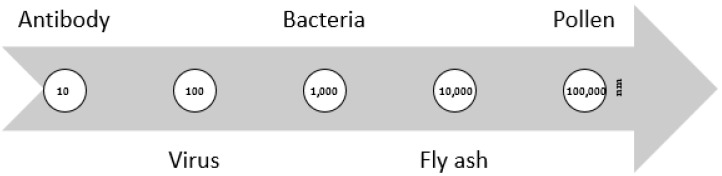
Different particle sizes to be filtered.

**Figure 2 materials-14-07147-f002:**
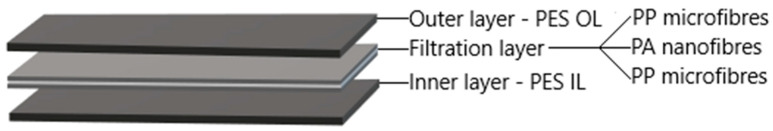
Multilayer system produced.

**Figure 3 materials-14-07147-f003:**
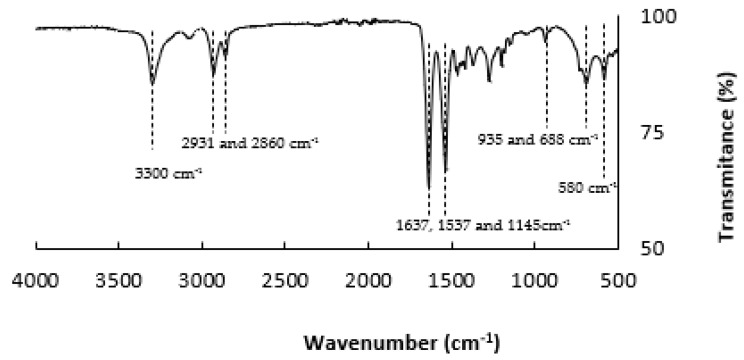
Fourier transform infrared spectra of electrospun PA nanofibres.

**Figure 4 materials-14-07147-f004:**
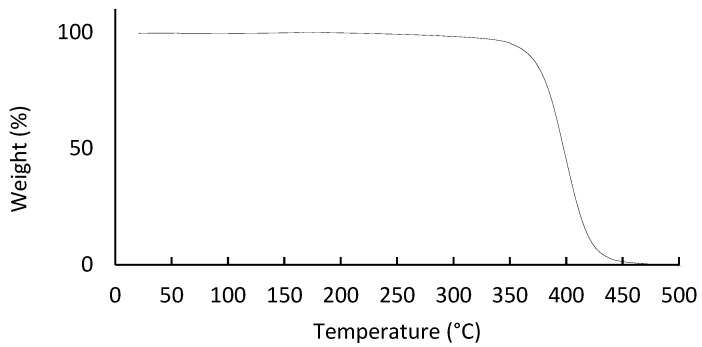
Thermogravimetric analysis curves of electrospun PA nanofibres.

**Figure 5 materials-14-07147-f005:**
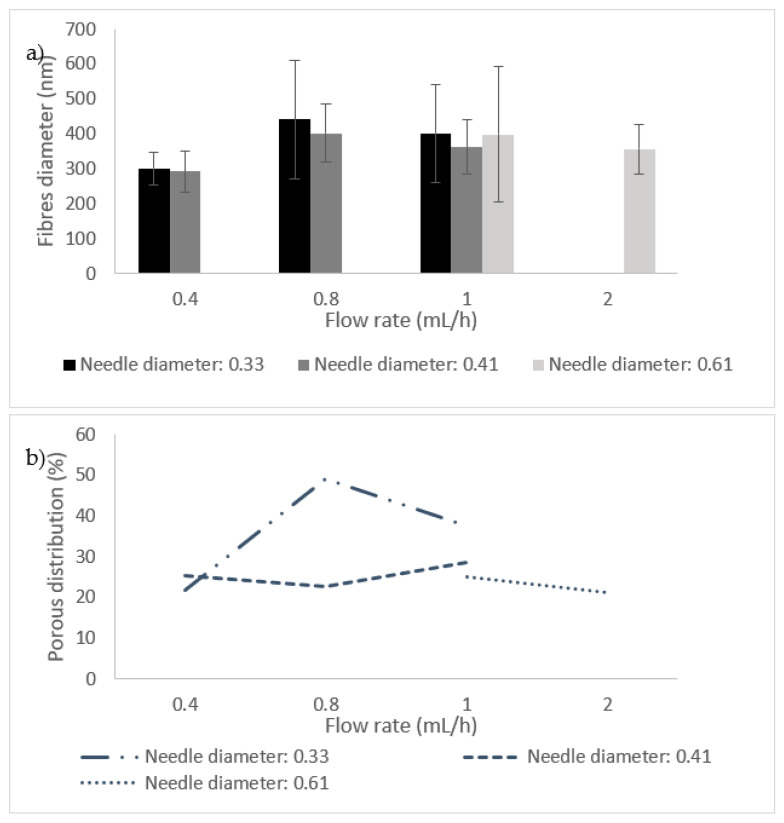
Influence of needle diameter (**a**) and flow rate (**b**) on fibre diameter and porous distribution values.

**Figure 6 materials-14-07147-f006:**
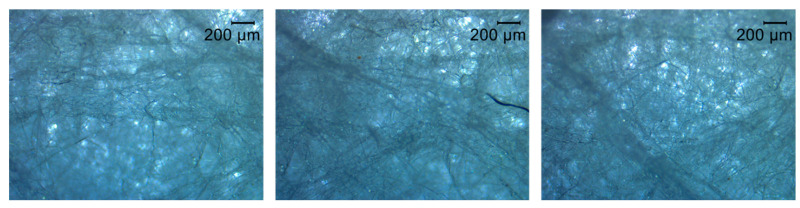
Morphology of (PP) microfibres under an optical microscope.

**Figure 7 materials-14-07147-f007:**
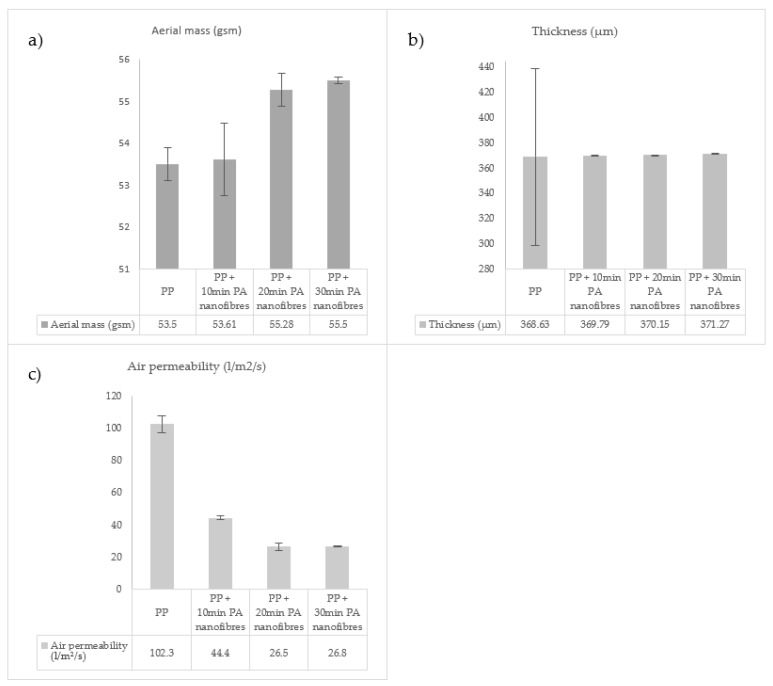
Aerial mass (**a**), thickness (**b**) and air permeability (**c**) of PP and PP + PA nanofibres combinations.

**Figure 8 materials-14-07147-f008:**
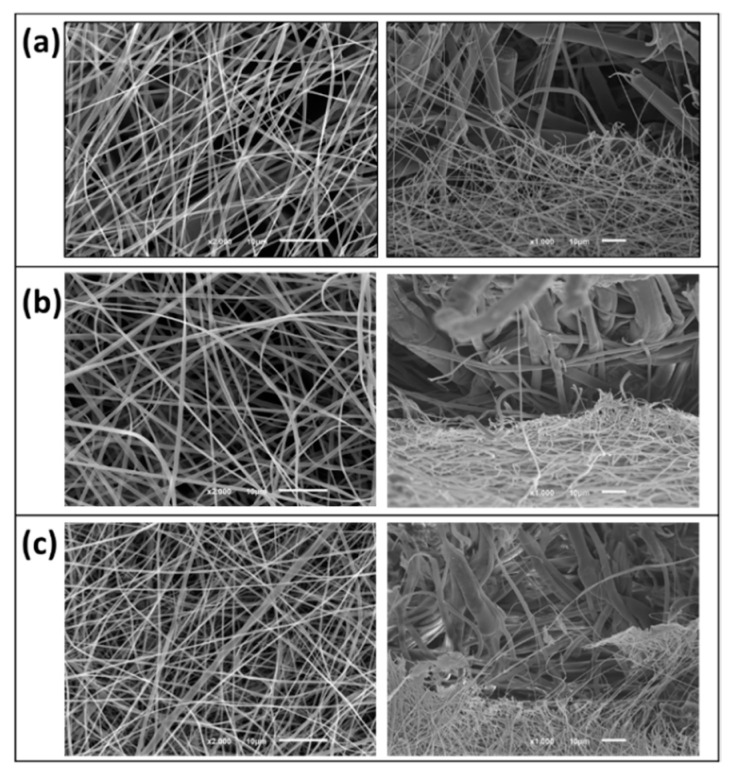
SEM images of the PP microfibres combined with PA nanofibres. The nanofibres were produced for (**a**) 10 min, (**b**) 20 min and (**c**) 30 min.

**Figure 9 materials-14-07147-f009:**
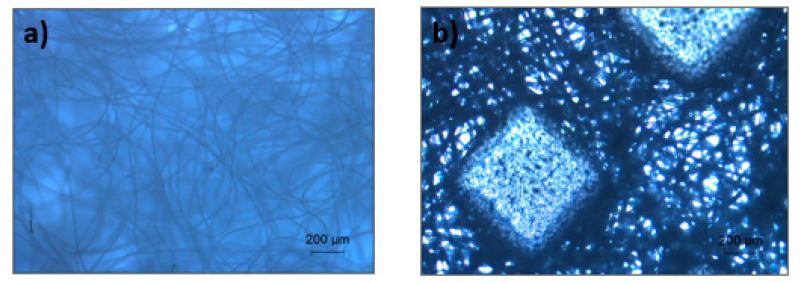
Morphology of PES nonwoven for (**a**) inner and (**b**) outer layer under an optical microscope.

**Table 1 materials-14-07147-t001:** Mechanisms of filtration and respective particle sizes to be filtered [7].

Mechanisms of Filtration	Size of Particles
Gravity sedimentation	Between 1 and 10 µm
Inertial impaction	Above 0.6 µm
Interception	Below 0.6 µm
Diffusion	Below 0.2 µm
Electrostatic attraction	Charged particles

**Table 2 materials-14-07147-t002:** Operational conditions tested during electrospinning production.

Sample	Solution Parameters		Electrospinning Parameters			
	Concentration (% (*w*/*v*))	Solvent	Voltage (kV)	Collector-Needle Distance (mm)	Flow Rate (mL/h)	Needle-Diameter (mm)
A	20	100% FA	28	100	0.4	0.33
B					0.8	
C					1	
D					0.4	0.41
E					0.8	
F					1	
G					1	0.61
H					2	
I	25				0.4	0.41

**Table 3 materials-14-07147-t003:** Multilayer systems tested in terms of filtration efficiency.

Reference	Inner Layer	Filtration Layer	Outer Layer	PA Deposition Time
Multilayer_0 min	PES IL	PP microfibres/PP microfibres (Note: without PA nanofibres)	PES OL	0 min
Multilayer _10 min		PP microfibres/PA (10 min)/PP microfibres		10 min
Multilayer _20 min		PP microfibres/PA nanofibres (20 min)/PP microfibres		20 min
Multilayer _30 min		PP microfibres/PA nanofibres (30 min)/PP microfibres		30 min

**Table 4 materials-14-07147-t004:** SEM images of the produced PA nanofibres and corresponding characteristics.

Sample and Production Parameters	SEM Images (×1000)	SEM Images (×5000)	Fiber Diameter ± STDEV (nm)	Porous Distribution (%)
A Flow rate = 0.4 mL/h Needle-diameter = 0.33 mm	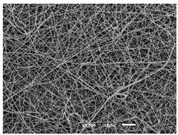	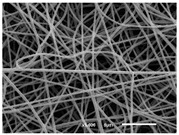	302 ± 46	21.66
B Flow rate = 0.8 mL/h Needle-diameter = 0.33 mm	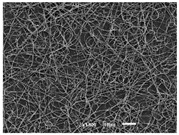	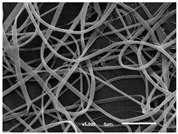	442 ± 170	48.97
C Flow rate = 1 mL/h Needle-diameter = 0.33 mm	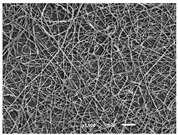	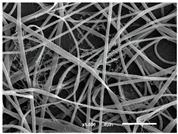	402 ± 141	37.50
D Flow rate = 0.4 mL/h Needle-diameter = 0.41 mm	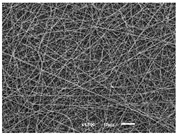	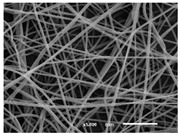	293 ± 60	25.27
E Flow rate = 0.8 mL/h Needle-diameter = 0.41 mm	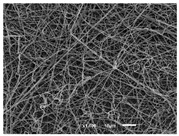	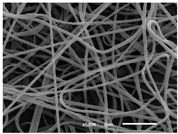	402 ± 84	22.71
F Flow rate = 1 mL/h Needle-diameter = 0.41 mm	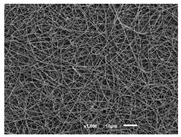	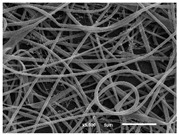	363 ± 80	28.57
G Flow rate = 1 mL/h Needle-diameter = 0.61 mm	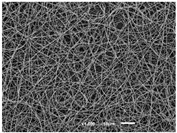	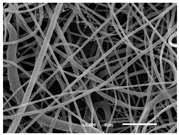	399 ± 193	25.09
H Flow rate = 2 mL/h Needle-diameter = 0.61 mm	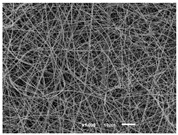	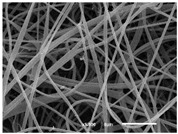	356 ± 70	21.11
I Flow rate = 0.4 mL/h Needle-diameter = 0.41 mm 25% (*w*/*v*) PA	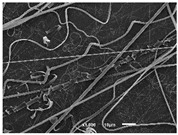	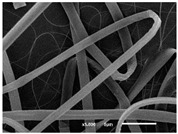	755 ± 711	56.05

**Table 5 materials-14-07147-t005:** PP microfibres characteristics.

Polymer	Average Fibres Diameter ± STDEV (µm)	Thickness (µm)	Air Permeability (L/m^2^/s)	Aerial Mass (gsm)
PP	3.7 ± 1.5	368.3	102.3	53.5

**Table 6 materials-14-07147-t006:** Filtration and respiratory evaluation of the four different multilayer systems studied.

Sample:	Multilayer _0 min	Multilayer _10 min	Multilayer _20 min	Multilayer _30 min
Test	Result			
Penetration of the filter material with sodium chloride after 3.5 min of exposure (%)	15.39	7.83	5.90	7.11
Respiratory resistance (mbar): Inhalation at 30 L/min	0.37	0.66	0.65	0.66
Respiratory resistance (mbar): Inhalation at 95 L/min	1.53	2.23	2.33	2.34
Respiratory resistance (mbar): Exhalation at 160 L/min	2.36	3.73	3.62	4.09

## Data Availability

Not applicable.

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
