# Peer review of "Study of the Filtration Performance of Multilayer and Multiscale Fibrous Structures"

_materials, 2021, doi:10.3390/ma14237147_

Round 1

Reviewer 1 Report

The manuscript entitled "Study of the filtration performance of multilayer and multi scale fibrous structures" is a significant study with impact on current concerns at international level, namely in filtration technology.

This article is well designed and demonstrates a systematic scientific approach and explanation of the observations. Nevertheless there should be no abbreviations in the abstract.

The Introduction section is well organized. Also, the following sections need not any re-organization, are well written with satisfactory scientific explanations, references and correlations.

Results are clearly described and analysed with high quality of images nevertheless it can be improved.

  • In Figure 1 and Table 1are badly visible text on a dark background.
  • In the line 117 is weight in unknown units (gsm), it should be explained.
  • In manuscript was found the minor errors (e. g. lines:143, 210, 272, 374, 427, 438)
  • FTIR and TGA analysis of PA nanofibers should be compared with some literature.
  • The legislation limits of the respiratory resistance should be written in paragraph and cited.

Conclusions are well written, providing useful information.

References have written in different font.

Based on the above concerns, I suggest a minor revision.

Reviewer 2 Report

In their «Study of the filtration performance of multilayer and multiscale fibrous structures», the authors report on the modification of a commercial PP meltblown fabric with a layer of electrospun PA nanofibers. After addition nonwoven topping and backing layers, filtration efficiency towards NaCl aerosol and differential pressure are accessed. Filtration efficiency was improved from 85 % to 94 % (15 % penetration to 6 % penetration) while the differential pressure increased from 1.53 to 2.33 mbar (best cases).

The degree of novelty is low, since (1) similar commercial products do already exist. (2) Electrospinning of PA is a standard process and (3) the applied testing methodology according to standard EN 149:2001+A1:2009 is not suited for fabrics. (4) The impact of the nanofiber layer on the filtration performance is modest and (5) finally, the authors waive the answer of questions raised in the introduction. Details are given below.

(1) L75: «To promote nanoparticles retention two factors should be considered: the use of fibres with very small diameters in the filter membrane and the application of a multi-layer system.» Many commercial filtration membranes are consituted following this exactly principle - e.g. SpureTex resporator with a penetration of 1.5 % at 0.5 mbar resistance, which consists also of 5 layers. (https://www.spur.cz/en/products/spurtex/spurtex-respirator/)

(2) L97: «Polyamide (PA) stands out as a potential polymer since it has excellent chemical stability and thermal resistance.» No reference to previous work on polyamide electrospinning is given. Are the here achieved fiber diameters by any means different from previous work? According to SciFinder, 830 references contain the concept "electrospinning + polyamide", the first one being the famous Reneker paper from 1996 (DOI:10.1088/0957-4484/7/3/009)

(3) L178: «filtration and respiratory evaluation were performed according to the standard EN 149:2001+A1:2009.» Standard EN 149:2001+A1:2009 "Filtering half masks to protect against particles" is not applicable to flat filtration media, but it is meant to test assembled face masks which may contain a multi-layer filtration medium as the one describe by the authors.

(4) L436: «Table 6: Filtration and respiratory evaluation of the four different multilayer systems studied.» A good way of comparing the combined effect of increased resporatory resistance with decreasing penetration is the filter quality factor (qf) (e.g. DOI:10.1080/02786826.2020.1829535) which is defined as qf = -ln(penetration)/delta(respiratory resistance). Here the four filters with and without nanofiber layer show qf values of  0.012, 0.011, 0.012, and 0.011 Pa-1. In other words, there is no significant improvement of the filter qf by adding the nanofiber layers.

(5) L50: «Figure 1: Different particle sizes to be filtered» In the introduction, the authors show a figure with a aerosol size range from 10 to 100000 nm and they mention different filtration mechanisms, which are size dependent. However, the presented results on NaCl filtration efficiency are size independent. Therefore, the proposed effect of the multilayer system (L75-L76) on the filtration efficiency for nanoparticles remains hypothetical.

Further minor remarks:

L92 «static charge» did the authors check the charge of their materials after electrospinning?

L105 «optimized to obtain fibres with very low diameters» literature evidence should be given to prove that 293 nm diameter for electrospun PA nanofibers is very low (which is not the case).

L136 «After optimizing the ideal conditions to produce the nanofibres, its deposition was performed over PP microfibres to set up a multilayer and multiscale system with higher performance in terms of small particles retention.» How did the authors achieve a homogeneous distribution of the PA nanofibers accross the whole surface of the PP microfibres? Did they use a moving needle head or a moving collector stage? Did the authors check homogeneity?

L148 «Figure 2: Multilayer system produced.» The description in Figure 2 does not correspond with the description in the text. PA-PP-PP vs PP-PA-PP (line 140)

L150 «Table 3: Multilayer systems …» PA deposition time for "Multilayer_30min" is missing

L240 «This result shows that to produce homogeneous mats of PA nanofibres with smaller diameters» Only a single experiment with 25 % w/v, but 8 experiments with 20 % w/v were performed. How to the authors know, that they could not find an other combination of parameters (distance, needle diameter, voltage) to produce small fibers even with 25 %? And what about 15 %? To find global optimum conditions, DOE should be considered.

L261 «Figure 5» Figure 5 is very confusing, in particular since the line-representation of the porous distribution does not correspond with the needle diameters of the bar plot representation (e.g. 0.33 plottet at 0.41 needle diameter bar). The content should be separated in two figures.

L327 «On one side, lower air permeability values may increase the filtration efficiency, however, on the other side, it may compromise the breathability parameter.» calculation of quality factor qf to account for effect of increased pressure drop with increasing efficiency is suggested.

Reviewer 3 Report

The manuscript of Pais et al describes the development of new materials by using the electrospinning for obtaining the fine fibers with different layers. As a results of this study a multilayer and multiscale system was developed comprising of a mesh of polyamide nanofibers including PES and Polypropylene microfibers. It was concluded that the addition of PA nanofibers enabled to achieve 10% higher filtration efficiency of the materials. The manuscript is clearly written and addresses an important problem of particle filtration that is especially actual topic during these times.

However, some language corrections are needed:

  1. Line 27: are lower than 2.5 µm. I suggest substituting lower with smaller.
  2. Line 29: “be together with virus or bacteria” needs rephrasing.
  3. Line 36:”the particle’s diameter is one of these properties” needs rephrasing.
  4. Line 45 please rephrase: “the decrease of the size of the fibres that compose the filter”.
  5. Line 66: “the electrostatic attraction can be related to particles of different sizes” needs clarification.

But please note I am not a native English speaker myself!

Figures and tables need also improvement:

Figure 2. Please improve the schematic representation of the multilayer system. It is not clear where are the different layers position in the final setup.

Figure 3. Please add label to each outstanding peak and especially to the ones that are referred to in the text.

Figure 4: Please add tick marks to the x and y axis for better understanding of the numeric values of the TGA analysis.

Table 4: please indicate if the “Fiber diameter” is ± STDEV or SEM

Line 248: “When analyzing the bars related to the diameter of the needles of 0.33 mm and 0.41 mm it is possible to conclude that when using a needle with a diameter equal to 0.41 mm nanofibres with lower diameters are always obtained.” Which bars are addressed?

Line 270: “From the microscopic images, it is possible to observe.” Rephrase: It is observed.

Table 5: please indicate  ± STDEV or SEM

Figure 7 should appear before the text and please improve the figure by adding letters on the same line etc.

Figure 8 should appear in the text before the content

“It was observed that the multilayer system without the presence of PA nanofibres achieved much higher values (15.39 %) than the systems with PA nanofibres (7.83, 5.9 and 7.11 %).” Please elaborate discussion, if the special properties of the PA could affect the material performance itself?

Last but not least, the authors pointed out that the 10% increase for air permeability was achieved but it was not clear how the 10% difference was calculated. Could you please refer to the calculation process.

Round 2

Reviewer 2 Report

In their revised version, the authors have adapted their manuscript following the minor comments from reviewer 1 and 3. They have also gone through the comments of reviewer 2 and adopted parts of the manuscript accordingly.

The authors have also added the sentence “It was observed that the penetration value of the multi-19 layer system with a PA membrane in the composition, produced for 20 min in the electrospinning, 20 is 2.7 times smaller than the penetration value of the system”. However, while they are emphasizing the improved filtration efficiency, the authors are HIDING the fact, that respiratory resistance has increased from 1.53 to 2.33 mbar or by a factor of 1.5. Therefore, it was suggested to introduce the filter quality factor, which accounts for both effects simultaneously since filtration efficiency and pressure drop are always a trade-off.

Still, the paper is not very novel and simply presents the current state of technical products. Among the already mentioned filter from SpureTex (https://www.spur.cz/en/products/spurtex/spurtex-respirator/) there are many other manufacturers using the here presented technology, e.g. NANO M.ON (https://obchod.nanomon.cz/premiova-nanovlakenna--zdravotnicka-maska-1ks/). Here, the authors fail to present their contribution beyond the current state. (In fact, even the Petryanov filters introduced in the early 80s contain nanofibers and show aerosol filtration efficiencies of > 99.99 %.)

Scientifically, it would have been more interesting to see, whether fractional filtration efficiency and most penetrating particle size would change by adding a layer of nanofibers. However, these data are not available through the applied experimental methodology (EN 149:2001+A1:2009).

Lack of novelty was also criticized in terms of the presented electrospinning procedure of polyamide. Electrospinning of polyamide is very well known and parameters influencing fiber properties have been well studied. If the authors spend several pages for describing their optimization process, they should compare their findings with those from the literature.

Finally, the paper is still written quite sloppy, e.g. mixing the “.” and “,” notation for decimal numbers.
